# Comparative Efficacy of 14-Day Tegoprazan-Based Triple vs. 10-Day Tegoprazan-Based Concomitant Therapy for *Helicobacter pylori* Eradication

**DOI:** 10.3390/jpm12111918

**Published:** 2022-11-17

**Authors:** Chan Hyuk Park, Myung Jin Song, Byung Wook Jung, Jung Ho Park, Yoon Suk Jung

**Affiliations:** 1Department of Internal Medicine, Hanyang University Guri Hospital, Hanyang University College of Medicine, Guri 11923, Republic of Korea; 2Division of Gastroenterology, Department of Internal Medicine, Kangbuk Samsung Hospital, Sungkyunkwan University School of Medicine, Seoul 03181, Republic of Korea

**Keywords:** *Helicobacter pylori*, eradication, P-CAB, tegoprazan, concomitant

## Abstract

Tegoprazan, a novel potassium-competitive acid blocker, is currently available for the treatment of *Helicobacter pylori* infection. We compared the efficacies of tegoprazan-based triple and concomitant therapies in a real-world practice. Data of patients treated with a 14-day tegoprazan-based triple therapy (50 mg of tegoprazan + 1000 mg of amoxicillin + 500 mg of clarithromycin twice daily) or 10-day tegoprazan-based concomitant therapy (50 mg of tegoprazan + 1000 mg of amoxicillin + 500 mg of clarithromycin + 500 mg of metronidazole twice daily) were retrospectively reviewed. Primary endpoint was eradication rate in the intention-to-treat (ITT) population. Of the 928 included patients, 551 and 377 were treated with triple and concomitant therapies, respectively. Eradication rate from ITT analysis was 76.4% (95% confidence interval [CI], 72.7–79.8%) in the triple therapy group and 85.9% (95% CI, 82.2–89.2%) in the concomitant therapy group (*p* < 0.001). Eradication rate in the per-protocol analysis was also higher in the concomitant therapy group than in the triple therapy group (triple vs. concomitant therapy: 84.5% [81.1–87.5%] vs. 91.1% [87.8–93.8%]). Overall adverse event rate was 29.0% in the triple therapy group and 45.9% in the concomitant therapy group (*p* < 0.001). Adherence rate was similar between the two groups (triple vs. concomitant therapy: 90.0 vs. 92.6%, *p* = 0.180). Overall, the 10-day tegoprazan-based concomitant therapy had superior efficacy than the 14-day tegoprazan-based triple therapy for *H. pylori* eradication. Although concomitant therapy showed common adverse events, adherence was comparable between the two therapies.

## 1. Introduction

Lamouliatte et al. first proposed triple therapy, consisting of a proton pump inhibitor (PPI), amoxicillin, and clarithromycin, for the treatment of *Helicobacter pylori* infection in 1993, and it is currently the most popular eradication regimen globally [1,2]. However, the efficacy of the conventional triple therapy regimen is decreasing owing to the resistance of *H. pylori* to clarithromycin [3,4,5,6]. Therefore, alternative regimens, including concomitant therapy (PPI + amoxicillin + clarithromycin + metronidazole) and bismuth-based quadruple therapy (PPI + bismuth + metronidazole + tetracycline), are recommended if the clarithromycin resistance rate is high, but clarithromycin susceptibility testing is unavailable [7,8].

According to Korean guidelines, 14-day conventional triple or 10-day concomitant therapy can be empirically chosen as the first-line treatment [8]. Potassium-competitive acid blockers (P-CABs), including vonoprazan, have been used to increase the eradication rate in conventional therapies [9]. In a meta-analysis, vonoprazan-based triple therapy showed a superior eradication rate compared to PPI-based triple therapy, especially in patients with clarithromycin-resistant *H. pylori* infection [10].

P-CABs have also been used in regimens other than triple therapy [11,12]. In a recent randomized controlled trial performed in the United States and Europe, P-CAB was effective in dual therapy (vonoprazan + amoxicillin) [11]. Although the eradication rate was similar between vonoprazan-based dual and PPI-based triple therapies in the overall study population, vonoprazan-based dual therapy was superior to PPI-based triple therapy in patients with clarithromycin-resistant *H. pylori* infection [11]. Tegoprazan, a novel P-CAB developed in Korea, is currently available for the treatment of *H. pylori* infection [13,14]. However, the efficacy of P-CAB-based concomitant therapy has not yet been investigated. Therefore, in this study, we aimed to compare the efficacy of two major P-CAB-based regimens (tegoprazan-based triple therapy vs. tegoprazan-based concomitant therapy) in a real-world practice.

## 2. Methods

### 2.1. Study Population

The data of patients aged ≥19 years who received tegoprazan-based triple or concomitant therapy for the treatment of *H. pylori* infection between March 2021 and May 2022 at Kangbuk Samsung Hospital and Hanyang University Guri Hospital in Korea were retrospectively reviewed. The exclusion criteria were as follows: (1) patients who previously received *H. pylori* eradication therapy, and (2) patients who had undergone subtotal gastrectomy. Demographics, symptoms, indications for *H. pylori* eradication therapy, medications, adverse events, and results of *H. pylori* eradication therapy were obtained. This study was approved by the Institutional Review Board of the Ethics Committee of each institution (Kangbuk Samsung Hospital, KBSMC 2022-06-034; Hanyang University Guri Hospital, GURI 2022-06-025). The need for informed consent was waived since this was a retrospective study.

### 2.2. H. pylori Eradication Regimen

The first-line treatment was employed with either tegoprazan-based triple (50 mg of tegoprazan + 1000 mg of amoxicillin + 500 mg of clarithromycin twice daily for 14 days) or concomitant therapy (50 mg of tegoprazan + 1000 mg of amoxicillin + 500 mg of clarithromycin + 500 mg of metronidazole twice daily for 10 days), according to the clinician’s preference. Successful eradication was assessed at least four weeks after treatment. If first-line *H. pylori* eradication therapy failed, second-line eradication therapy via bismuth-containing quadruple therapy was administered as follows: 50 mg of tegoprazan or 20 mg of rabeprazole twice daily in combination with 120 mg of bismuth four times a day, 500 mg of metronidazole three times a day, and 500 mg of tetracycline four times a day for 14 days.

### 2.3. Study Endpoint and Measurements

The first-line *H. pylori* eradication rate in the intention-to-treat (ITT) analysis was the primary endpoint. Secondary endpoints included the first-line eradication rate in the per-protocol (PP) analysis, second-line eradication rate in ITT and PP analyses, and adverse events associated with *H. pylori* eradication therapy.

*H. pylori* infection was evaluated using one or more of the following tests: ^13^C-urea breath test, rapid urease test, and histologic examination with modified Giemsa staining. *H. pylori* eradication was determined to have failed if any of the above three tests were found positive.

Atrophic gastritis was visually assessed based on the Kimura–Takemoto classification [15]. Drug adherence was defined as the administration of ≥80% of prescribed medications. The severity of adverse events was classified as mild (transient symptoms that improved spontaneously), moderate (symptoms that required management), and severe (symptoms that led to an emergency visit) [14].

### 2.4. Statistical Analysis

Continuous variables, including age and body mass index (BMI), were compared using Student’s *t*-test. Categorical variables were compared using Fisher’s exact test. In ITT analysis, patients who received insufficient medication (<80% of the prescribed medications) or were lost to follow-up were considered to have failed the treatment. For PP analysis, patients with insufficient medications and those lost to follow-up were excluded. Factors associated with the failure of first-line *H. pylori* eradication therapy were identified using logistic regression analysis. Variables with *p*-values < 0.1 in the univariable logistic analysis were included as covariates in the multivariable logistic regression model. *p* < 0.05 was considered significant for group comparisons. All statistical analyses were performed using the R (version 4.0.4; R Foundation for Statistical Computing, Vienna, Austria).

## 3. Results

### 3.1. Study Population and Baseline Patient Characteristics

A total of 965 patients who were treated with tegoprazan-based triple or concomitant therapy for *H. pylori* infection were included in this study (Figure 1). Of these, 34 patients who had received a previous *H. pylori* eradication therapy and three who had undergone subtotal gastrectomy were excluded. As a result, a total of 928 patients were included in the study. In the ITT population, 551 patients treated with a 14-day tegoprazan-based triple therapy and 377 patients treated with a 10-day tegoprazan-based concomitant therapy were analyzed. In the PP population, 496 and 349 patients in the triple and concomitant therapy groups, respectively, remained after excluding 18 patients with insufficient medication and 65 patients lost to follow-up. Ninety-two patients who failed to treat the *H. pylori* infection received second-line eradication therapy.

Table 1 shows the baseline patient characteristics. Mean age was 55.7 ± 11.1 years in the 14-day tegoprazan-based triple therapy group and 55.3 ± 11.1 years in the 10-day tegoprazan-based concomitant therapy group (*p* = 0.539). The proportion of males was 55.0% in the triple therapy group and 45.4% in the concomitant therapy group (*p* = 0.004). There were no significant differences in other baseline characteristics, including BMI, smoking status, alcohol consumption, and comorbidities.

Patient symptoms and endoscopic findings are presented in Table 2. The most common symptom in both groups was abdominal discomfort; however, most patients showed no symptoms. *H. pylori*-associated gastritis was the most common indication for eradication therapy. Atrophic gastritis was common in both groups (triple vs. concomitant therapy: 79.1 vs. 75.9%, *p* = 0.002).

### 3.2. Efficacy of the First-Line Eradication Therapy

The eradication rate of the first-line therapy is shown in Figure 2. In the ITT population, the 10-day tegoprazan-based concomitant therapy showed a superior eradication rate compared to the 14-day tegoprazan-based triple therapy (triple vs. concomitant therapy: 76.4% [95% confidence interval [CI], 72.7–79.8%] vs. 85.9% [95% CI, 82.2–89.2%], *p* < 0.001). In the PP population, the eradication rate was also higher in the concomitant therapy group than in the triple therapy group (triple vs. concomitant therapy: 84.5% [95% CI, 81.1–87.5%] vs. 91.1% [95% CI, 87.8–93.8%], *p* = 0.004).

A total of 92 patients, in whom the first-line *H. pylori* eradication therapy failed, received the second-line therapy. Of them, 21 patients were treated with a 14-day rabeprazole-based bismuth-containing quadruple therapy, while 71 patients were treated with a 14-day tegoprazan-based bismuth-containing quadruple therapy. Significant differences in the eradication rates were not identified in either ITT or PP analyses (ITT analysis: rabeprazole-based quadruple therapy, 95.2% [95% CI, 79.8–99.5%] vs. tegoprazan-based quadruple therapy, 85.9% [95% CI, 76.5–92.5%], *p* = 0.443; PP analysis: rabeprazole-based quadruple therapy, 100.0% [95% CI, 87.1–100.0%] vs. tegoprazan-based quadruple therapy, 96.7% [95% CI, 89.7–99.3%], *p* > 0.999) (Appendix A). Additionally, we evaluated the eradication rate of second-line therapy according to the type of first-line eradication regimen. The second-line eradication rate in patients who had received the 14-day triple therapy as first-line therapy was 85.1% (95% CI, 75.8–91.8%) in ITT analysis and 96.7% (95% CI, 89.9–99.3%) in PP analysis. In patients who had received 10-day concomitant therapy as first-line therapy, the second-line eradication rate was 96.7% (95% CI, 89.9–99.3%) in ITT analysis and 100.0% (95% CI, 86.5–100.0%) in PP analysis. The second-line eradication rate did not differ between the first-line eradication regimens (*p* = 0.114 in ITT analysis and *p* > 0.999 in PP analysis).

### 3.3. Adherence and Adverse Events

The adherence and adverse events of the first-line *H. pylori* eradication therapy are shown in Table 3. The adherence rate was 90.0% in the triple therapy group and 92.6% in the concomitant therapy group (*p* = 0.180). The rate of loss to follow-up was higher in the triple therapy group than in the concomitant therapy group (triple vs. concomitant: 8.5% vs. 4.8%, *p* = 0.028). However, the rate of insufficient medication did not differ between the groups (triple vs. concomitant therapy: 1.5% vs. 2.7%, *p* = 0.193). The rate of overall adverse events was higher in the concomitant therapy group than in the triple therapy group (triple vs. concomitant therapy: 29.0% vs. 45.9%, *p* < 0.001). Although most adverse events were mild, 1.3% of patients in the triple therapy group and 2.4% of patients in the concomitant therapy group discontinued their medications owing to adverse events. One patient (0.3%) in the concomitant therapy group visited the emergency center because of nausea, vomiting, diarrhea, and general weakness. The patient fully recovered with supportive care, including intravenous hydration. The incidence of dysgeusia, the most common adverse event in the triple therapy group, did not differ between the triple and concomitant therapy groups (triple vs. concomitant therapy: 11.3 vs. 11.1%, *p* = 0.958). The most common adverse event in the concomitant therapy group was diarrhea, which was more common in the concomitant therapy group than in the triple therapy group (triple vs. concomitant therapy: 9.3 vs. 22.8%, *p* < 0.001).

In the second-line therapy, the adherence rate was 85.7% for rabeprazole-based bismuth-containing quadruple therapy and 84.5% for tegoprazan-based bismuth-containing quadruple therapy (*p* > 0.999) (Appendix A). The overall adverse event rate was 61.9% in the rabeprazole-based therapy group and 45.1% in the tegoprazan-based therapy group (*p* = 0.175).

### 3.4. Factors Associated with H. pylori Eradication Failure

Logistic regression model for the failure of first-line therapy is shown in Table 4. Univariate analysis showed that triple therapy (vs. concomitant therapy), non-adherence, and diabetes were associated with treatment failure. Under adjusting confounding variables, triple therapy (vs. concomitant therapy, odds ratio [OR] [95% CI] = 2.08 [1.34–3.24]), non-adherence (vs. adherence, OR [95% CI] = 4.35 [1.31–14.4]), female sex (vs. male, OR [95% CI] = 1.59 [1.06–2.39]), and diabetes (vs. non-diabetes, OR [95% CI] = 1.90 [1.08–3.34]) were significantly associated with treatment failure.

## 4. Discussion

The present study demonstrated that the 10-day tegoprazan-based concomitant therapy was superior than the 14-day tegoprazan-based triple therapy group for the successful eradication of *H. pylori*. Eradication rate of PPI-based concomitant therapy has been reported to be higher than that of the PPI-based triple therapy [16,17]. Our study supports the beneficial effects of concomitant therapy, even with tegoprazan-based eradication regimens. The 10-day concomitant therapy requires three different antibiotics (amoxicillin, clarithromycin, and metronidazole); however, it can shorten the treatment duration compared to the 14-day triple therapy.

Another notable finding of our study was that the 10-day concomitant therapy reached an acceptable rate for *H. pylori* eradication therapy in South Korea, where the clarithromycin resistance rate of *H. pylori* is high. Generally, an acceptable eradication regimen should have an eradication rate ≥ 85% in ITT analysis and ≥90% in PP analysis [18]. Although 14-day triple therapy is one of the recommended regimens in Korea [8], neither PPI- nor P-CAB-based triple therapy showed acceptable eradication rates in our previous study [19]. In that study, the eradication rate of the 14-day rabeprazole-based triple therapy was 75.4% in ITT analysis and 83.5% in PP analysis [19]. Additionally, the eradication rate of the 14-day tegoprazan-based triple therapy was 76.7% in ITT analysis and 83.4% in PP analysis [19]. Even though tegoprazan is used for *H. pylori* eradication therapy, a concomitant therapy regimen may be required to ensure the sufficient eradication if *H. pylori* infection is treated without knowing the antibiotic susceptibility in regions with high clarithromycin resistance. Although the efficacy of tegoprazan-based concomitant therapy was analyzed for the first time in our study, the comparative efficacy between PPI-based triple and concomitant therapies has been examined in many studies [16,17]. In a recent worldwide network meta-analysis involving 68 randomized-controlled trials, PPI-based concomitant therapy showed superior eradication efficacy compared to PPI-based triple therapy in both the West and East Asia (concomitant vs. triple: West, OR 2.39 [95% CI, 1.15–4.94]; East Asia, OR 2.01 [95% CI, 1.41–2.87]) [17]. In South Korea, where the clarithromycin resistance rate is >15%, the eradication rate of PPI-based concomitant therapy was also higher than that of PPI-based triple therapy (concomitant vs. triple: OR 1.72 [95% CI, 1.17–2.53]) [16].

Although we showed the superior eradication rate of the concomitant therapy, one potential concern of first-line concomitant therapy is the overlap of metronidazole between the first-line and second-line eradication therapies when bismuth-containing quadruple therapy is used as a rescue therapy. Given that metronidazole is included in the concomitant therapy regimen and that the metronidazole resistance rate is high in South Korea [20], the role of metronidazole in second-line therapy (bismuth-containing quadruple regimen) may be unclear. However, in our study, the eradication rate of second-line therapy did not differ between the first-line eradication regimens (bismuth-containing quadruple therapy following triple vs. concomitant therapy). Although only a small number of patients were included in the analysis for second-line therapy, we could not identify the inferior efficacy of second-line therapy in patients who had received concomitant therapy as the first-line treatment for *H. pylori* infection. Because the typical dosage of metronidazole is higher in bismuth-containing quadruple therapy (500 mg thrice a day) than in concomitant therapy (500 mg twice a day), metronidazole resistance may be overcome in some cases during second-line eradication therapy. The effects of first-line eradication regimens on second-line eradication therapy should be investigated in future large-scale studies.

Another concern associated with concomitant therapy is the high rate of adverse events. In our study, approximately half of the patients in the concomitant therapy group experienced at least one adverse event. High adverse event rates may have affected the discontinuation of medications. However, only a minor proportion of patients experienced moderate-to-severe adverse events. In most cases, adverse events were unlikely to affect the medication compliance. The proportion of patients with insufficient medication was also similar between triple and concomitant therapies. The relatively short treatment duration in the concomitant therapy group may be associated with a comparable proportion of patients with sufficient medication because the 10-day concomitant therapy requires ≥8 d medication, whereas the 14-day triple therapy requires >11 d medication to meet the criterion for sufficient medication.

Our logistic regression model showed that triple therapy, compared to concomitant therapy, was independently associated with *H. pylori* eradication failure. Non-adherence, female sex, and diabetes were independent risk factors for eradication failure. Although the reason for the different eradication rates between males and females is unclear, several studies have reported an association between female sex and failed eradication [21,22,23]. Another study in Korea showed that A2143G mutated *H. pylori*, which associated with clarithromycin resistance, was more common in females than in males [24]. The association between diabetes and eradication failure has not yet been fully elucidated. However, several hypotheses have been proposed that diabetes contributes to a poor eradication rate. Glycosylation prevents drug binding (antibiotics) in the blood [25]. Impaired microvasculature of the gastric mucosa in patients with diabetes and diabetic gastroparesis may reduce the bioavailability of antibiotics [26,27,28]. Additionally, frequent use of antibiotics in patients with diabetes may affect the antibiotic resistance [29].

Although we first evaluated the efficacy of P-CAB-based concomitant therapy, our study has several limitations. First, this is a retrospective study. Therefore, successful eradication was not evaluated in some patients lost to follow-up. However, we can understand real-world practice in patients who received tegoprazan-based triple or concomitant therapies in the current study. Although patients receiving tegoprazan-based concomitant therapy experienced more adverse events than those receiving tegoprazan-based triple therapy, their adherence rate was >90%. Second, the data on antibiotic resistance were unavailable. Therefore, we could not analyze the efficacy of eradication therapy according to the antibiotic resistance status. However, our real-world study on tegoprazan-based therapy could be performed based only on an empirical treatment setting because triple therapy is no longer allowed if clarithromycin resistance is identified [8]. Third, most patients in the study showed no symptoms and the most common indication for *H. pylori* eradication was *H. pylori*-associated gastritis. In South Korea, many patients diagnosed with *H. pylori* infection are treated to reduce the risk of gastric cancer development, even though they do not have any symptoms or *H. pylori*-related diseases. Therefore, caution should be exercised when generalizing the findings of our study. In addition, since our study was conducted in Koreans with high rates of *H. pylori* infection and clarithromycin resistance, there are limitations in generalizing our results to other countries.

## 5. Conclusions

Despite these limitations, we can understand the real-world efficacy of the 10-day tegoprazan-based concomitant therapy for the treatment of *H. pylori* infection in regions where clarithromycin resistance rate is high. The 10-day tegoprazan-based concomitant therapy was superior than the 14-day tegoprazan-based triple therapy for *H. pylori* eradication. Although approximately half of the patients receiving the 10-day tegoprazan-based concomitant therapy experienced adverse events, most adverse events were mild and acceptable.

## Figures and Tables

**Figure 1 jpm-12-01918-f001:**
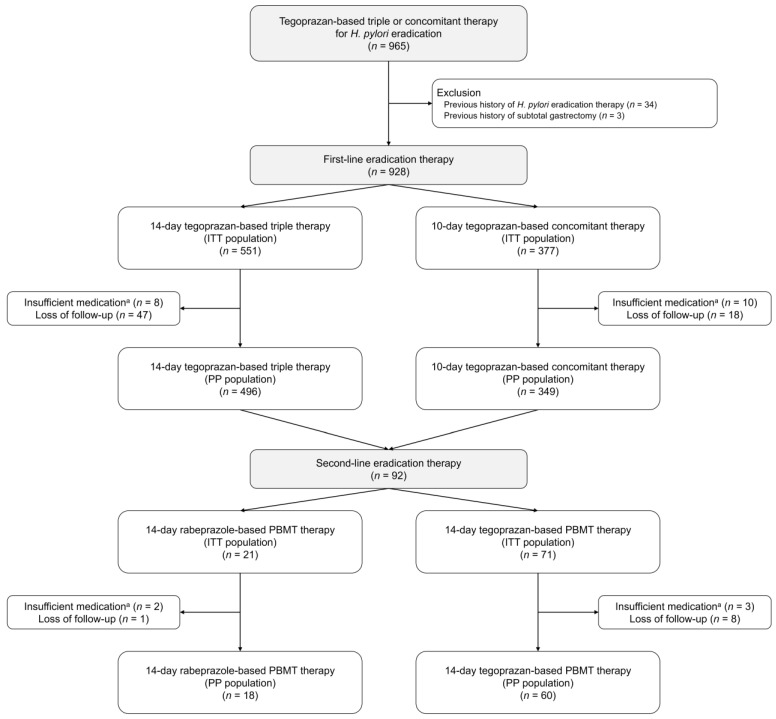
Study flow diagram. ^a^ Insufficient medication is determined as administration of <80% of prescribed medication. PBMT indicates bismuth-containing quadruple therapy comprising a proton pump inhibitor (PPI; or potassium-competitive acid blocker [P-CAB]), bismuth, metronidazole, and tetracycline. ITT, intention-to-treat; PP, per-protocol.

**Figure 2 jpm-12-01918-f002:**
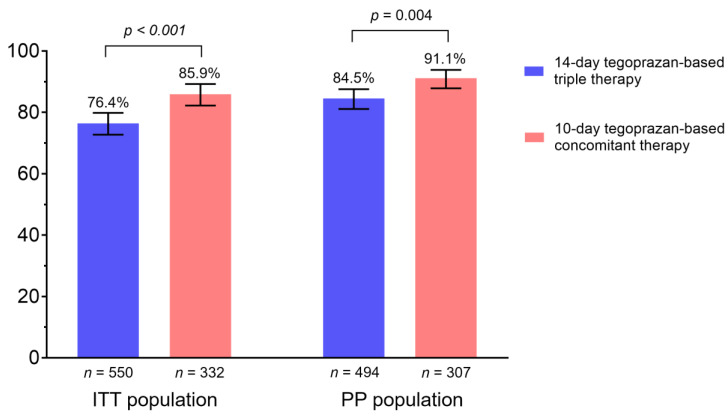
Success rate of the first-line *Helicobacter pylori* eradication therapy. ITT, intention-to-treat; PP, per-protocol.

**Table 1 jpm-12-01918-t001:** Baseline characteristics of included patients.

	14-Day Tegoprazan-Based Triple Therapy (*n* = 551)	10-Day Tegoprazan-Based Concomitant Therapy (*n* = 377)	*p*-Value
**Age, year, mean ± SD**	55.7 ± 11.1	55.3 ± 11.1	0.539
**Male, n (%)**	303 (55.0)	171 (45.4)	0.004
**BMI, ^a^ kg/m, mean ± SD**	24.3 ± 3.3	24.0 ± 3.2	0.163
**Smoking habit**			0.712
Never smoker	306 (55.5)	216 (57.3)	
Former smoker	146 (26.5)	101 (26.8)	
Current smoker	99 (18.0)	60 (15.9)	
**Alcohol use**			0.419 ^b^
Absent	289 (52.5)	184 (48.8)	
Present	262 (47.5)	193 (51.2)	
<2/week	157 (28.5)	109 (28.9)	
≥2/week	105 (19.1)	84 (22.3)	
**Comorbidity**			
Hypertension	120 (21.8)	94 (24.9)	0.262
Cardiovascular disease	24 (4.4)	14 (3.7)	0.628
Diabetes	60 (10.9)	39 (10.3)	0.792
Cerebrovascular accident	14 (2.5)	10 (2.7)	0.916

^a^ BMI is missing in one patient in the triple therapy group. ^b^ *p*-value of alcohol use was calculated with reference to alcohol non-use between the two groups. BMI, body mass index; SD, standard deviation.

**Table 2 jpm-12-01918-t002:** Baseline symptoms and endoscopic findings.

	14-Day Tegoprazan-Based Triple Therapy (*n* = 551)	10-Day Tegoprazan-Based Concomitant Therapy (*n* = 377)	*p*-Value
**Symptom**			
Reflux symptom ^a^	12 (2.2)	10 (2.7)	0.641
Nausea or vomiting	7 (1.3)	5 (1.3)	>0.999
Gastric soreness	44 (8.0)	48 (12.7)	0.017
Abdominal discomfort	79 (14.3)	61 (16.2)	0.441
Abdominal pain	16 (2.9)	13 (3.4)	0.640
Others ^b^	15 (2.7)	23 (6.1)	0.011
**Indication for *H. pylori* eradication**			0.010
Gastric and duodenal ulcers	2 (0.4)	4 (1.1)	
Gastric ulcer	37 (6.7)	11 (2.9)	
Duodenal ulcer	43 (7.8)	34 (9.0)	
MALT lymphoma	1 (0.2)	0 (0.0)	
EGC treated with ESD	10 (1.8)	1 (0.3)	
Gastric adenoma treated with ESD	7 (1.3)	2 (0.5)	
*H. pylori*-associated gastritis	451 (81.9)	325 (86.2)	
**Nodular gastritis ^c^**	46 (8.3)	33 (8.8)	0.828
**Atrophic gastritis ^c,d^**			0.002
Absent (C-0)	115 (20.9)	53 (14.1)	
Present			
C-1	127 (23.1)	80 (21.2)	
C-2	74 (13.5)	48 (12.7)	
C-3	93 (16.9)	75 (19.9)	
O-1	62 (11.3)	71 (18.8)	
O-2	48 (8.7)	40 (10.6)	
O-3	30 (5.5)	10 (2.7)	

^a^ Reflux symptoms include heartburn and acid regurgitation. ^b^ Other symptoms include globus sensation, anorexia, and belching. ^c^ There are two missing values for nodular gastritis and atrophic gastritis in the triple therapy group. ^d^ Severity of atrophic gastritis is determined by Kimura-Takemoto classification. MALT, muocsa-associated lymphoid tissue; EGC, early gastric cancer; ESD, endoscopic submucosal dissection.

**Table 3 jpm-12-01918-t003:** Adherence and adverse events of first-line *H. pylori* eradication therapy.

	14-Day Tegoprazan-Based Triple Therapy (*n* = 551)	10-Day Tegoprazan-Based Concomitant Therapy (*n* = 377)	*p*-Value
**Adherence, ^a^ *n* (%)**	496 (90.0)	349 (92.6)	0.180
Loss of follow-up	47 (8.5)	18 (4.8)	0.028
Insufficient medication	8 (1.5)	10 (2.7)	0.193
**Adverse event, ^b^ *n* (%)**			
Any adverse event	160 (29.0)	173 (45.9)	<0.001
Mild	153 (27.8)	164 (43.5)	
Moderate	7 (1.3)	8 (2.1)	
Severe	0 (0.0)	1 (0.3)	
General weakness	1 (0.2)	7 (1.9)	0.009
Dizziness	1 (0.2)	7 (1.9)	0.009
Hedache	2 (0.4)	4 (1.1)	0.231
Myalgia	0 (0.0)	0 (0.0)	N/A ^d^
Acid regurgitation	1 (0.2)	3 (0.8)	0.310
Nausea or vomiting	26 (4.7)	34 (9.0)	0.009
Dysgeusia	62 (11.3)	42 (11.1)	0.958
Abdominal discomfort	6 (1.1)	6 (1.6)	0.561
Abdominal pain	3 (0.5)	4 (1.1)	0.451
Diarrhea	51 (9.3)	86 (22.8)	<0.001
Constipation	3 (0.5)	1 (0.3)	0.650
Skin rash	9 (1.6)	3 (0.8)	0.379
Others ^c^	5 (0.9)	14 (3.7)	0.003

^a^ Adherence is determined as administration of ≥80% of prescribed medications. ^b^ Percentage is calculated based on the ITT population. ^c^ Other adverse events include insomnia, dry mouth, and sores on the tongue. ^d^ N/A indicates that *p*-value cannot be calculated because the number of events in both groups is zero. N/A, not applicable.

**Table 4 jpm-12-01918-t004:** Factors associated with failure of first-line *H. pylori* eradication ^a^.

	*n*	Failure *n* (%)	Univariable Analysis		Multivariable Analysis	
			OR (95% CI)	*p*-Value	OR (95% CI)	*p*-Value
**Treatment duration**						
14-day tegoprazan-based triple	501	81 (16.2)	1.5 (1.27–3.01)	0.003	2.08 (1.34–3.24)	0.001
10-day tegoprazan-based concomitant	356	32 (9.0)	1		1	
**Adherence**						
Adherent	844	108 (12.8)	1		1	
Non-adherent	13	5 (38.5)	4.26 (1.67–13.26)	0.012	4.35 (1.31–14.4)	0.016
**Age**						
<60 years	519	65 (12.5)	1			
≥60 years	338	48 (14.2)	1.16 (0.77–1.73)	0.478		
**Sex**						
Male	433	48 (11.1)	1		1	
Female	424	65 (15.3)	1.45 (0.97–2.17)	0.067	1.59 (1.06–2.39)	0.026
**BMI**						
<25 kg/m^2^	537	68 (12.7)	1			
≥25 kg/m^2^	320	45 (14.1)	1.13 (0.75–1.69)	0.558		
**Smoking habit**						
Never smoker	481	70 (14.6)	1			
Former smoker	233	29 (12.4)	0.84 (0.53–1.33)	0.445		
Current smoker	143	14 (9.8)	0.64 (0.35–1.17)	0.146		
**Alcohol use**						
Absent	437	62 (14.2)	1			
<2/week	246	33 (13.4)	0.94 (0.60–1.48)	0.779		
≥2/week	174	18 (10.3)	0.70 (0.40–1.22)	0.206		
**Comorbidity**						
Hypertension	194	30 (15.5)	1.28 (0.81–2.01)	0.287		
Cardiovascular disease	34	7 (20.6)	1.75 (0.75–4.13)	0.198		
Diabetes	91	19 (20.9)	1.89 (1.09–3.27)	0.024	1.90 (1.08–3.34)	0.027
Cerebrovascular accident	24	1 (4.2)	0.28 (0.04–2.09)	0.215		
**Symptom**						
Reflux symptom ^b^	19	1 (5.3)	0.36 (0.05–2.72)	0.323		
Nausea or vomiting	12	0 (0.0)	N/A	0.999		
Gastric soreness	87	14 (16.1)	1.30 (0.71–2.39)	0.399		
Abdominal discomfort	128	18 (14.1)	1.09 (0.63–1.88)	0.751		
Abdominal pain	28	6 (21.4)	1.84 (0.73–4.64)	0.196		
Others ^c^	36	6 (16.7)	1.34 (0.54–3.28)	0.530		
**Indication for *H. pylori* eradication**						
Peptic ulcer	118	14 (11.9)	0.89 (0.49–1.63)	0.713		
MALT lymphoma	1	1 (100.0)	N/A	>0.999		
EGC treated with ESD	11	3 (27.3)	2.49 (0.65–9.55)	0.184		
Gastric adenoma treated with ESD	9	1 (11.1)	0.83 (0.10–6.71)	0.861		
*H. pylori*-associated gastritis	718	94 (13.1)	1			
**Nodular gastritis**	72	8 (11.1)	0.81 (0.38–1.74)	0.587		
**Atrophic gastritis ^d^**						
Normal (C-0)	148	19 (12.8)	1			
Mild (C-1, C-2)	302	34 (11.3)	0.86 (0.74–1.57)	0.626		
Moderate (C-3, O-1)	284	42 (14.8)	1.18 (0.66–2.11)	0.581		
Severe (O-2, O-3)	122	18 (14.8)	1.18 (0.59–2.35)	0.649		

^a^ This analysis is performed on participants who received a follow-up test for *H. pylori* eradication. ^b^ Reflux symptoms include heartburn and acid regurgitation. ^c^ Other symptoms include insomnia, dry mouth, and sores on the tongue. ^d^ Severity of atrophic gastritis is determined by Kimura-Takemoto classification. MALT, mucosa-associated lymphoid tissue; OR, odds ratio; CI, confidence interval; N/A, not applicable.

## Data Availability

The data presented in this study are available on request from the corresponding author.

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
