# Peer review of "Comparative Efficacy of 14-Day Tegoprazan-Based Triple vs. 10-Day Tegoprazan-Based Concomitant Therapy for Helicobacter pylori Eradication"

_jpm, 2022, doi:10.3390/jpm12111918_

Round 1

Reviewer 1 Report

The authors compared Helicobacter pylori eradication treatment cases in Korea using tegoprazan, a novel P-CAB, in combination with triple therapy and concomitant therapy.

The results showed that although the concomitant therapy caused slightly more adverse events, most of them were not severe, and the therapy was superior in eradicating Helicobacter pylori.

The study design is well organized, and the manuscript is expected to be the subject of future meta-analyses.

The words "Helicobacter pylori" in the abstract and in the reference should be italicized. Also, "helicobacter pylori" in the references should be capitalized. If this is not an editorial specification, please correct it.

What is the difference between nothing is written and "N/A" in tables? I would like to see a correction or an explanation that I can understand.

I think "Variable" in the table is unnecessary. It may be a matter of taste, so correction is not essential.

The items in the table are very difficult to read. This is also a matter of taste, but I prefer left-alignment rather than center-alignment. Each item and the words that summarize it are also shown in the same style, and there is room for improvement. For example, "Symptom" should be bolded in table2, and the related items below should not be bolded. It is also very difficult to read the items such as "Gastric and duodenal ulcers" where the items are spread over two lines in the table, etc. It is necessary to simplify the notation of the items of triple therapy and concomitant therapy, or to make a new line so that the left-most item fits in one line. 

Reviewer 2 Report

This retrospective study by Park and coll. offer a viewpoint of the efficacy of novel anti-secretive agents for the treatment of H. pylori infection.

In my opinion the text and tables should be shortened  to make the manuscript more readable.

Furthermore, the authors should compare their results with similar studies from different countries, also of the surrounding geographic areas, to offer a more complete view of the problem. A brief discussion. or a figure, regarding the entity of CLA resistance can be useful for this purpose.

In addition, a conclusion section must be defined.

At last, in Tab. 1 delete the duplicate "P-CAB, potassium-competitive acid blocker" in the caption.

Round 2

Reviewer 2 Report

The authors' revision of the the manuscript and their replies are satisfactory, and overall quality of it has been improved.